# StackEval: Benchmarking LLMs in Coding Assistance

**Nidhish Shah** *
Prosus AI
nidhish.shah@prosus.com

**Zulkuf Genc** *
Prosus AI
zulkuf.genc@prosus.com

**Dogu Araci** *
Prosus AI
dogu.araci@prosus.com

## Abstract

We present two comprehensive benchmarks to evaluate the performance of language models in coding assistance tasks, covering code writing, debugging, code review, and conceptual understanding. Our main contribution includes two curated datasets: StackEval, a large-scale benchmark derived from Stack Overflow questions, and StackUnseen, a dynamic benchmark featuring the most recent Stack Overflow content. These benchmarks offer novel insights into the capabilities and limitations of LLMs, particularly in handling new and emerging content. Additionally, we assess LLMs' proficiency as judges for coding tasks using a curated, human-annotated dataset, exploring their evaluation capabilities and potential biases, including whether they favor their own generated solutions. Our findings underscore the potential of these benchmarks to advance LLM development and application in coding assistance. To ensure reproducibility, we publicly share our datasets and evaluation code at `https://github.com/ProsusAI/stack-eval`.

## 1 Introduction

Language Models have become indispensable tools for software developers, significantly helping with tasks such as solution implementation, troubleshooting and code reviewing. Over 40% of developers leverage AI to boost efficiency, reduce errors, optimize code, and facilitate learning [24]. Despite their widespread adoption, there remains a critical need for systematic evaluation to fully understand and optimize LLM performance across these diverse coding assistance tasks.

Creating high-quality datasets that cover a wide range of programming languages and coding assistance tasks demands significant resources. Evaluating these tasks is particularly challenging due to their open-ended nature, requiring intensive effort from highly skilled domain experts. Moreover, the risk of benchmark leakage or exposure bias, where test data might unintentionally be included in the training sets of models, adds another layer of complexity to the evaluation process [16].

To address these challenges, we introduce a comprehensive suite of benchmarks and evaluation methodologies designed to thoroughly assess LLMs in coding assistance. Our key contributions are:

1. **StackEval: A Comprehensive Multi-Language, Multi-Task Coding Benchmark**. This comprehensive benchmark comprises meticulously curated Stack Overflow questions covering 25 programming languages and four task types — debugging, implementation, optimization, and conceptual understanding. By leveraging human-verified solutions, StackEval enables a thorough assessment of LLM capabilities across a wide range of coding tasks.

2. **StackUnseen: A Dynamic Benchmark for Emergent Coding Challenges**. This dataset is a companion to our main coding assistance benchmark, specifically created from the latest Stack Overflow data dump. While the main dataset covers historical content, StackUnseen focuses on recent and emergent programming questions. Continuously updated, it evaluates LLM performance on new technologies and evolving coding practices, addressing the

---

*Equal contribution.

challenges posed by rapidly changing programming landscapes and the absence of up-to-date information in LLM training sets. This benchmark mitigates the issue of data leakage, enabling a more accurate assessment of LLM capabilities on unseen coding queries.

3. **A Comprehensive Study on LLMs-as-Judges for Coding Tasks**: We present the first extensive study evaluating LLMs as judges for coding assistance tasks. Leveraging our StackEval benchmark, we curate a dataset of LLM-generated answers evaluated by human domain experts. We investigate how techniques such as inclusion of reference answers and chain-of-thought prompting affect evaluation accuracy, and employ statistical analyses to examine potential biases, particularly whether LLM judges show favoritism towards their own generated solutions. This comprehensive approach led to a robust evaluation methodology for coding assistance tasks, achieving an 84.4% success rate in judging the acceptability of generated responses.

To further the advancement of AI coding assistance, we are making our StackEval and StackUnseen datasets publicly available. The StackUnseen dataset will be updated periodically to maintain its relevance, allowing for the assessment of LLMs' adaptability to emerging challenges. Furthermore, we also share an interactive leaderboard interface, enabling users to explore specific LLM performances on various tasks, such as "C++ advanced debugging" questions.

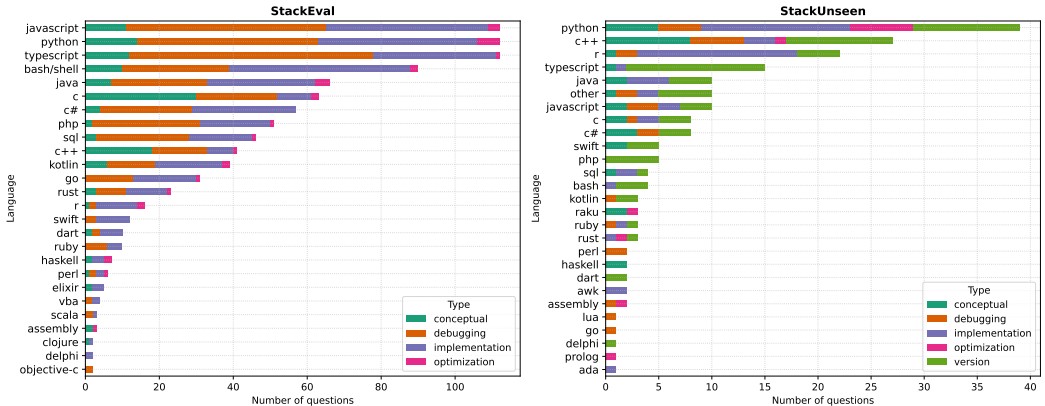

Figure 1: **StackEval & StackUnseen Programming Language Distribution.** The questions are subdivided based on the programming languages and type. The distribution of languages is sampled based on popularity of said languages as indicated in the Stack Overflow Developer Survey, 2023 [24].

## 2 Related Works

**Coding Benchmarks.** HumanEval [9], the most widely recognized coding benchmark, consists of 164 Python coding problems designed to evaluate the problem-solving abilities of language models. While useful, its small size and focus on a single programming language limit its applicability for comprehensive coding assistance evaluation. It has also been public since 2021, which likely led to its inclusion in the training sets of recent LLMs. SWE-BENCH [15] provides a thorough benchmark for evaluating language models on real-world software engineering tasks, utilizing 2,294 GitHub issues and pull requests to test models' ability to edit extensive codebases. This benchmark reveals significant limitations in current state-of-the-art models, which struggle with the complexity and scope of real-world code editing tasks. However, its reliance on Python repositories and GitHub issues may limit its generalizability across different programming languages and coding assistant tasks. MultiPL-E [8] extends existing benchmarks by translating HumanEval and MBPP Python coding problems into 18 additional programming languages, evaluating language models' code generation capabilities across diverse languages. It uses adapted unit tests to assess both syntactical correctness and functional accuracy. However, it does not evaluate models' performance in broader coding assistant tasks beyond code generation, such as debugging and code review. HumanEval-X [42] and MBXP [5] are multilingual extensions of the existing HumanEval benchmark, translating

Python problems into multiple languages. Although these benchmarks cover more languages than the original HumanEval, they remain limited to function completion tasks.

**Evaluation Methodologies for LLMs**. Evaluating the performance of LLMs, especially in open-ended tasks, presents significant challenges. Traditional metrics such as BLEU [26] and ROUGE [19] are often inadequate for assessing code correctness and quality [9, 6, 28], emphasizing the need for more sophisticated evaluation methods. This has led to the development and adoption of benchmark suites that include unit tests, which provide a more direct and meaningful assessment of code functionality and correctness. However, these traditional metrics and unit tests often fall short when evaluating open-ended coding assistance tasks that blend code and natural language. These tasks demand an understanding of context, intent, and the nuances of human language, alongside technical code accuracy. This complexity requires more sophisticated and context-aware evaluation methodologies to effectively assess the performance of LLMs in these hybrid scenarios.

Recent approaches [41, 17, 18, 27] have explored using language models for automated evaluation, leveraging their capabilities to assess the quality and relevance of generated outputs. Frameworks like MT-Bench and ChatBot Arena utilize LLMs to judge responses, simulating a peer-review process that aligns well with human preferences, achieving over 80% agreement. These auto-evaluation frameworks often rely on pairwise comparisons and win rates. While this approach can enhance the robustness of evaluations, particularly when a strong reference is lacking, it primarily indicates which model performs better relative to others without providing an absolute measure of task performance. Additionally, pairwise comparisons are susceptible to position and verbosity biases [13, 17, 36].

To address these limitations, we developed a specific evaluation metric and rubric to quantify responses based on the task's objectives. This approach offers a clearer understanding of a model's effectiveness for the task at hand. Furthermore, we incorporated validated ground truth answers into the auto-evaluation process, which helps the evaluator focus on the relevant information, thereby minimizing confusion from long text and mitigating position bias.

**LLMs in Code Optimization.** The application of language models to code optimization tasks has emerged as a promising area of research. [10] introduced LLM Compiler, a model trained on 546 billion tokens of LLVM-IR and assembly code. This approach achieved 77% of the optimizing potential of an autotuning search for code size reduction. Similarly, [30] developed an open-source solution using CodeGen, a billion-parameter LLM, reporting a 2.5x speedup for over 25% of processed code. These studies demonstrate the potential of LLMs in code optimization tasks, opening up new avenues for research into their effectiveness across different optimization problems and their integration with traditional compiler techniques.

## 3 Dataset Curation

### 3.1 Collection and Filtration

**Composition.** In curating our datasets for benchmarking LLMs in coding assistance, we started with a comprehensive corpus derived from the public dumps of multiple Stack Exchange platforms, such as Stack Overflow, Ask Ubuntu, Super User, Unix & Linux, Software Engineering and Server Fault. These sources, while rich in content, often contain a heterogeneous mix of high and low quality material. To ensure the inclusion of only the most relevant and high-quality questions, we adopted a thorough filtration process.

**Filtration.** The selection criteria required each question to have at least one upvote, and an accepted answer that also received at least one upvote, ensuring community validation of both the question's relevance and the answer's correctness. We excluded question-answer pairs containing images or links, which can complicate text-based processing, and those exceeding 16,000 characters to maintain content conciseness. Following the initial filtration, we leveraged Stack Exchange tags to identify the programming language in each question. We then sampled the questions for each language based on its popularity as indicated by the Stack Overflow Developer Survey, 2023 [24], reflecting real-world usage patterns.

**Annotation.** Each question was further annotated using GPT-4 Turbo [22] to classify both the question type and complexity level. The complexity levels were categorized as *Beginner*, *Intermediate*, and *Advanced*, providing a structured framework to assess the LLMs' performance across varying degrees of difficulty. Additionally, questions were tagged according to five distinct types pertinent to coding

tasks, which are essential for the StackEval and StackUnseen benchmarks. It's important to note that these tags are suggestive and were not used for sampling the questions; they serve primarily for analysis purposes. The question types include: (i) *Conceptual*: Questions aimed at gaining a deeper understanding of programming concepts, such as "How does the yield keyword work in Python?"; (ii) *Debugging*: Questions that require resolving specific programming errors, for example, "Why am I getting a syntax error in this code?"; (iii) *Implementation*: Questions about executing specific programming tasks, often necessitating code snippets in responses, such as "How can I reverse a string in JavaScript?"; (iv) *Optimization*: Questions focused on improving the efficiency of existing code or soliciting code reviews, like "How can I optimize this algorithm for better performance?"; and (v) *Version* (only relevant to StackUnseen): Questions that require knowledge about specific (latest) versions of software, for instance, "How does the new `except*` statement work in Python 3.11?".

The final step in our dataset preparation involved a meticulous manual review. This review served to confirm the dataset's effectiveness as a valid evaluation tool, focusing on three key aspects: clarity of questions, relevance to real-world coding scenarios, and technical accuracy. These criteria are crucial for accurately assessing the capabilities of LLMs in coding-related tasks.

### 3.2 Benchmark Suite

We propose two comprehensive benchmarks, StackEval, and StackUnseen, designed to evaluate LLMs' performance in coding-related tasks. Figure 1 illustrates the final distribution of questions across various programming languages and tasks. Additionally, we describe the LLM-as-a-Judge benchmark, which plays a crucial role in evaluating LLMs' judgment capabilities.

**StackEval Benchmark** is designed to evaluate the combined natural language understanding and coding capabilities of LLMs, reflecting the complex nature of real-world coding assistance tasks. It includes a diverse set of 925 questions, sampled from January 2018 to September 2023, ensuring a comprehensive coverage across various programming languages, question types, and complexity levels. This dataset's granular nature allows for detailed insights into the specific capabilities and limitations of different LLMs, facilitating a deeper understanding of their performance across a spectrum of coding tasks.

**StackUnseen Benchmark** addresses the dynamic nature of coding practices and the continuous evolution of programming languages. The first release of StackUnseen covered questions from September 2023 to March 2024, and now has been expanded to include an additional set from March to May 2024. This benchmark is updated semi-annually with new questions, ensuring that the dataset remains relevant and challenging, reflecting the latest trends in software development. The inclusion of recent questions helps prevent potential test-train leakage and allows for a more current evaluation of LLM capabilities. Regular updates allow for longitudinal studies of LLM performance, providing insights into how these models adapt to new data and whether they exhibit any learning or adaptation over time.

**LLM-as-a-Judge Benchmark** consists of a carefully curated set of 136 questions from the StackEval benchmark, chosen to ensure a representative mix of complexities and topics. For each question, an answer is generated using an LLM (referred to as LLM-$x$ in Figure 2, which is one of GPT-4 Turbo [22], GPT-3.5 Turbo [7], CodeLlama-34B [29] or Mistral Medium [14]). This LLM-generated answer is then evaluated by human domain experts, who compare it to the original accepted Stack Overflow answer based on the evaluation criteria outlined in Table 1. A third domain expert then verified these annotations to ensure consistency and accuracy. This multi-layered annotation process is designed to provide a robust dataset for evaluating LLMs as judges in coding-related tasks.

## 4 Evaluation Methodology

The evaluation of coding assistance tasks presents unique challenges due to their open-ended nature. Unlike traditional benchmarks with predefined correct answers, coding problems often have multiple valid solutions, making objective assessment complex. To address this challenge, we adopted the LLM-as-a-Judge [41] framework for evaluation, which allows for an assessment that takes into account the complexity and variability inherent in programming queries. Our evaluation framework incorporates four key components, (i) *Question*: The original question from Stack Overflow; (ii)

*Reference Answer*: The accepted answer with the highest votes on Stack Overflow, serving as a benchmark for quality and relevance; (iii) *LLM-generated Answer*: The solution produced by the language model being evaluated; and (iv) *Chain-of-Thought Reasoning*: A step-by-step analysis process encouraged in the model's evaluation [37].

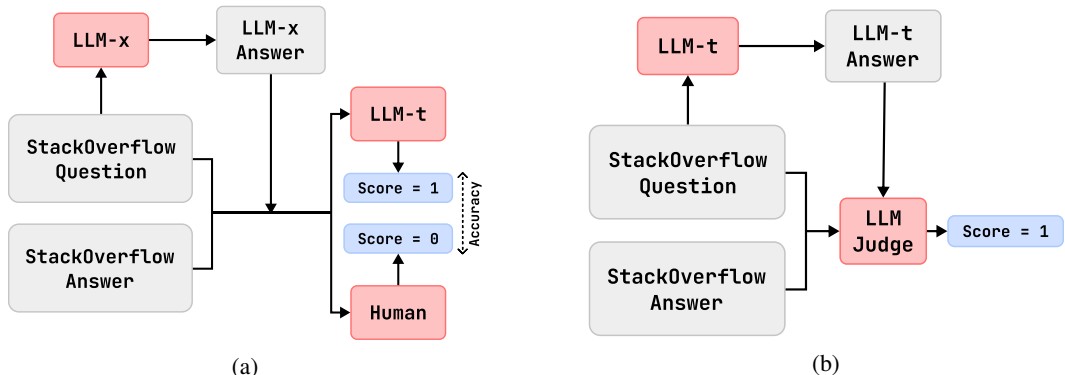

(a)                 (b)

Figure 2: **Evaluation methodology for assessing LLMs on coding tasks. a)** LLM-as-a-Judge benchmark (CoT + Ref. Answer) comparing LLM-$t$ (model under test) against human experts when evaluating answers from LLM-$x$ models . **b)** Coding assistance evaluation where LLM-$t$ generates StackOverflow answers, scored by an LLM judge.

To standardize the evaluation process, we propose the *acceptance score* metric, inspired by the concept of accepted answers on Stack Overflow. This metric assesses generated responses across three critical dimensions: accuracy, completeness, and relevance. For a response to be deemed acceptable, it must excel in all three dimensions; any deficiency leading the user to continue searching for a solution renders the response unacceptable. We conducted our evaluation in two phases, each focusing on different aspects of the model's performance. A summary of both phases is found in Figure 2.

| Category | Score | Description |
|---|---|---|
| Optimal | 3 | The answer is highly accurate and detailed, providing comprehensive and useful guidance, enhancing the user's understanding and application. |
| Acceptable | 2 | The answer is accurate and relevant, covering the main points sufficiently, enabling the user to proceed without additional help. |
| Partially Unacceptable | 1 | The answer has some correct information but significant inaccuracies or lacks important details, necessitating additional research. |
| Fully Unacceptable | 0 | The answer is incorrect, irrelevant, or contains significant errors and misinformation, requiring further search by the user. |

Table 1: Evaluation Rubric for LLM-Generated Answers in Coding Assistance Tasks.

**LLM-as-a-Judge**. In this phase, we assess the model's ability to evaluate coding solutions accurately. We compared several evaluation methodologies, instructing the model to output a score based on the evaluation rubric in Table 1, with different components provided as follows:

1. **Question and Answer Only**: The model received only the question and the answer to be evaluated, without any additional context or guidance.

2. **Question, Answer, and Reference**: The model received the question, the answer to be evaluated, and a reference answer (the highest-voted accepted answer from Stack Overflow). This configuration allowed the model to compare the given answer against a high-quality benchmark.

3. **Question and Answer with Chain-of-Thought**: The model received the question and answer, with instructions to analyze the question's requirements and examine the answer's accuracy, completeness, and relevance before scoring, potentially leading to more considered evaluations.

4. **Question, Answer, Reference, and Chain-of-Thought**: The model received all elements and was instructed to use chain-of-thought reasoning, combining reference comparison with structured analysis.

We report the accuracy against human labelers' acceptance scores (binarized based on acceptability). The accuracy metric reflects how well the LLMs' evaluations align with those of human experts. This metric was chosen to quantify how well the model can discern between acceptable and unacceptable coding solutions.

**Coding Assistance**. For this phase, we evaluate the model's ability to generate high-quality coding solutions. We report the acceptance rate, i.e, the proportion of model-generated solutions deemed acceptable, derived from acceptance scores of 2 or 3 based on the evaluation rubric. This approach ensures that generated responses are not only technically accurate but also practically applicable and contextually appropriate.

## 5 Experiments

In this section, we first present the experimental analysis of various LLMs as judges for coding assistance tasks, exploring the impact of different prompt configurations, such as the inclusion of reference answers and chain-of-thought reasoning. Following this, we evaluate the performance of these LLMs using the StackEval and StackUnseen benchmarks. Finally, we conduct a statistical analysis to study the phenomenon of favoritism in LLMs [25], where LLMs tend to favor their own generations.

### 5.1 LLM-as-a-Judge Benchmark

Like any automated evaluation system, using LLMs as judges raises important questions about reliability and consistency. These models might struggle with nuanced technical assessments or exhibit biases from their training on specific programming paradigms, potentially affecting the fairness of their evaluations. To mitigate these concerns, we verify the LLM judges' evaluations using human annotations.

The LLM-as-Judge benchmark consists of tuples comprising a Stack Overflow question, an LLM-generated solution, and where applicable, the accepted Stack Overflow answer. These elements were used to format the judging prompt as outlined in Section 4. Response generation was configured with a near-deterministic temperature setting ($T = 0.01$) and fixed seed ($s = 42$). We then evaluated the models' alignment with human judgment by comparing their acceptance scores against human-annotated ratings using an accuracy metric. We report the mean evaluation accuracy across 10 runs in Table 2.

| Model | Baseline | CoT | Ref. Answer | CoT + Ref. Answer |
|---|---|---|---|---|
| GPT-4 Turbo | 78.3% ± 1.9% | 73.9% ± 1.2% | 83.6% ± 0.4% | **84.4% ± 1.8%** |
| Claude-3.5 Sonnet | 81.3% ± 0.3% | 81.7% ± 0.6% | 82.1% ± 1.3% | **82.8% ± 0.2%** |
| WizardLM-2 8x22B | 80.7% ± 0.2% | 77.6% ± 0.2% | **81.9% ± 1.6%** | 81.5% ± 0.3% |
| Llama3.1-70B | 78.7% ± 0.4% | 76.1% ± 0.4% | 80.3% ± 0.3% | **81.2% ± 0.9%** |
| Mistral Large 2 | 79.9% ± 1.2% | 80.1% ± 1.7% | **81.1% ± 0.5%** | 80.5% ± 0.2% |
| Gemini-1.5 Pro | 71.3% ± 1.1% | 66.6% ± 1.9% | 72.9% ± 0.3% | **74.9% ± 1.5%** |

Table 2: **LLM-as-a-Judge Benchmark.** Mean accuracy ± 95% confidence interval for selected LLMs [22, 3, 38, 12, 1, 33] evaluated across different prompt configurations with and without Chain-of-Thought (CoT) reasoning and access to the reference answer.

Our experiments reveal several key patterns in LLM judging capabilities. The inclusion of reference answers consistently improves evaluation accuracy across all models tested, suggesting that

comparison-based assessment is more reliable than standalone evaluation. Interestingly, Chain-of-Thought (CoT) reasoning alone does not enhance performance and in some cases slightly degrades it, indicating that structured reasoning without proper context might lead to overthinking or misanalysis. The combination of CoT reasoning with reference answers yields the best overall performance, though this improvement is marginal compared to using reference answers alone. This suggests that while structured reasoning can be beneficial, access to high-quality reference solutions is the more crucial component for accurate evaluation.

## 5.2 StackEval and StackUnseen Benchmarks

In evaluating LLMs on the StackEval and StackUnseen benchmarks, we presented each LLM with the coding questions directly, without supplementary prompting or instructions. Response generation was configured with a near-deterministic temperature setting ($T = 0.01$) and fixed seed ($s = 42$), with outputs limited to 2048 tokens. We then assessed these responses using the LLM-as-a-Judge framework, employing the evaluation prompt detailed in Figure 8. The acceptance rates across various LLMs from different providers are presented in Table 3.

| Model | Provider | StackEval | StackUnseen |
|---|---|---|---|
| O1 Preview | OpenAI | **95.5%** | **83.0%** |
| Claude-3.5 Sonnet | Anthropic | 89.5% | 76.3% |
| Gemini-1.5 Pro | Google | 90.7% | 71.6% |
| Llama3.1-70B Nemotron | Nvidia | 86.9% | 66.5% |
| Mistral Large 2 | Mistral | 81.9% | 52.6% |
| WizardLM-2 8x22B | Microsoft | 80.2% | 50.5% |
| Llama3.1-405B | Meta | 76.5% | 47.9% |

Table 3: **The StackEval and StackUnseen Benchmarks.** The acceptance rate of performance of various LLMs [21, 3, 33, 20, 1, 12] on the coding assistance benchmarks. (See Table 5 in Appendix for full results.)

The O1 Preview model, which incorporates reasoning-chains [21], achieves the highest performance across benchmarks. This aligns with findings from recent studies [31, 40, 39]. Notably, open-source models like Llama3.1-70B Nemotron and WizardLM-2 8x22B are competitive with proprietary offerings, indicating a narrowing gap between open-source and commercial LLMs.

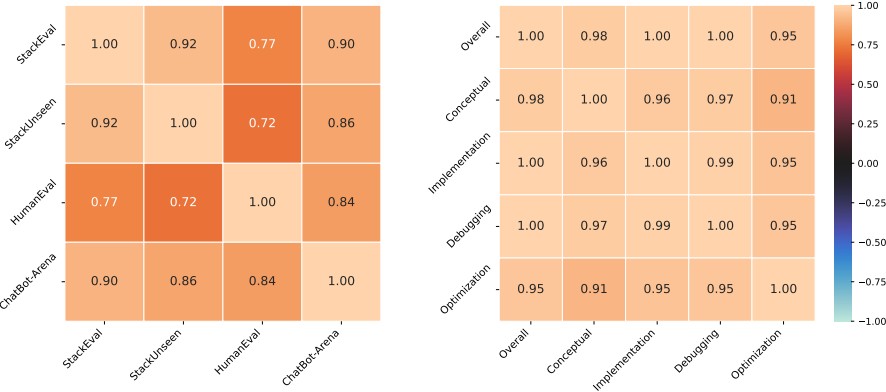

Figure 3: *Left:* Correlation between model performance across different coding benchmarks shows strong positive correlations (0.72-0.92). *Right:* Model performance across different question types within the benchmark are very highly correlated (0.91-1.00), suggesting consistent performance across task categories.

Analysis of the StackEval and StackUnseen benchmarks reveals consistent model rankings (Table 5), suggesting stable performance characteristics across both historical and emerging content. However, we observe a significant decrease in acceptance rates on unseen content. Figure 4 shows that

higher-capacity models and those using reasoning-chains (the same models that excel in StackEval) demonstrate better generalization to novel problems. This suggests that the capabilities that drive strong performance on established problems also contribute to better handling of unseen challenges.

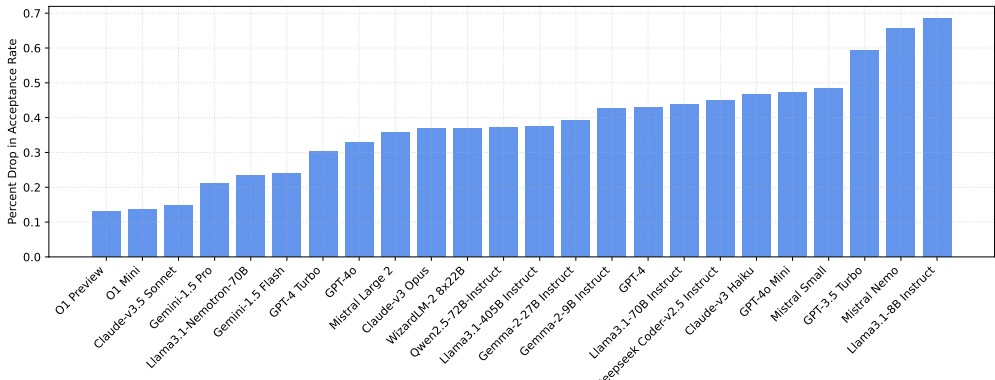

Figure 4: **Model Performance Degradation on Recent Problems.** LLMs with higher StackEval scores show smaller acceptance rate drops on StackUnseen, suggesting better generalization to contemporary problems.

## 5.3 Self-Preference in LLM Judges

Previous work has identified potential biases in LLM-based evaluation systems, particularly the phenomenon of self-preference where models may preferentially score their own generations. While such biases have been demonstrated in subjective tasks like summarization [32, 25], no systematic study has examined self-preference in code evaluation contexts.

We analyze how LLM judges evaluate their own code generations compared to solutions from other models, in the presence and absence of reference answers. Our study focuses on coding tasks, which typically have at least one correct solution, providing an opportunity to examine whether objectively correct solutions influence self-preference behavior.

| Model | Baseline | CoT + Ref. Answer |
|---|---|---|
| GPT-4 Turbo | 0.218 | 0.326 |
| Claude-3.5 Sonnet | 0.139 | 0.187 |
| WizardLM-2 8x22B | 0.224 | 0.156 |
| Llama3.1-70B | **0.048** | 0.279 |
| Mistral Large 2 | **0.039** | 0.163 |
| Gemini-1.5 Pro | 0.198 | 0.251 |

Table 4: **Self-Preference Analysis Results.** P-values from Wilcoxon Signed Rank Test comparing self-scores vs other-scores. **Bold** indicates $p < 0.05$ where we reject $H_0$ in favor of $H_1$, suggesting presence of self-preference bias.

Using the models detailed in Table 2, we generated and evaluated responses across all models. Consistent with previous experiments, we maintained temperature $T = 0.01$ and a fixed seed ($s = 42$). Each model evaluated both its own solutions and those generated by other models. To analyze the presence of self-preference, we conducted the Wilcoxon Signed Rank Test to study the following hypotheses, where self-scores ($S_i$) represent the score a model $m$ assigns to its own solutions, and other-scores ($S_o$) represent the average scores assigned to model $m$ by all other models for the same questions.

$H_0$ : The median difference between self-scores ($S_i$) and other-scores ($S_o$) is zero.

$H_1$ : The median difference between self-scores ($S_i$) and other-scores ($S_o$) is greater than zero.

Contrary to expectations from previous studies, in the presence of reference answers, we consistently fail to reject $H_0$ across all models (all $p > 0.05$). Without reference answers, we fail to reject $H_0$ in four out of six cases, with marginal rejections ($p = 0.048$ and $p = 0.039$) in the remaining two. These findings suggest that reference answers may serve as an objective anchor that eliminates self-preference bias, while even without such anchors, the bias appears minimal in coding tasks. These results indicate that in domains where objective correctness criteria exist, such as programming tasks, LLM-based evaluation systems may be less susceptible to self-preference biases than in more subjective domains. The results of the statistical tests are summarized in Table 4.

# 6 Discussion

## 6.1 Dataset Curation

Stack Overflow, while an invaluable resource for programmers, harbors inherent biases stemming from its community guidelines and voting system. The platform's emphasis on practical, specific questions tends to favor immediately applicable solutions, potentially skewing the data towards implementation-focused content. Strict moderation policies often lead to the exclusion of subjective, exploratory, or unconventional questions, thereby narrowing the scope of topics represented. Moreover, the community's preference for concise, code-centric answers further biases responses towards pragmatic solutions. Popular technologies and programming languages typically receive more attention and higher-quality answers, resulting in an uneven representation across different areas of programming knowledge. The voting system amplifies this effect by promoting content that aligns with mainstream interests, potentially overshadowing niche but valuable contributions.

Our benchmark, derived from Stack Overflow data, inherits these characteristics, reflecting a diverse range of real-world programming questions and solutions. While it excels in representing practical, community-vetted programming knowledge, it may not fully capture an LLM's capabilities across all programming paradigms or theoretical aspects. Researchers should consider these limitations when interpreting benchmark results and complement this evaluation with additional methods to gain a comprehensive understanding of LLM performance in coding tasks.

## 6.2 Evaluation Methodology

Our approach of employing LLMs as judges for evaluating coding assistance capabilities offers significant advantages in efficiency and scalability. Despite our comprehensive study into the accuracy and self-preference of LLM judges, it is essential to acknowledge inherent limitations. These judges may be subject to the same constraints as the models they evaluate, potentially leading to assessment blind spots, particularly in highly specialized or complex coding tasks. Moreover, LLM judges might exhibit biases towards certain response styles prevalent in their training data (which are harder to detect and quantify), potentially skewing evaluations regardless of solution quality.

Our methodology focuses on one-shot model predictions, which provides a standardized approach for assessment; it does not account for LLM-based agents capable of running and verifying code. Additionally, our evaluation criteria primarily assess the correctness and relevance of code solutions and may not fully capture crucial aspects of real-world software development such as code efficiency, maintainability, or adherence to best practices. These factors, though critical, present significant challenges for consistent evaluation using current LLM capabilities.

Furthermore, our LLM-as-Judge benchmark, although comprehensive, is based on a static snapshot of Stack Overflow questions. This temporal limitation could potentially impact the long-term effectiveness of the judges, as their performance may vary with newer or different types of questions. Consequently, continuous updating and re-evaluation of LLM judges are necessary to ensure their ongoing effectiveness and relevance in the rapidly evolving field of software development.

# 7 Conclusion

In this study, we introduce two comprehensive benchmarks, StackEval and StackUnseen, to evaluate the effectiveness of LLMs in coding assistance tasks across a variety of programming languages. These benchmarks comprehensively assess capabilities in code writing, debugging, code review, and

answering conceptual questions, utilizing a diverse set of coding questions from Stack Overflow. Furthermore, we conduct an extensive investigation into the use of LLMs as judges for coding tasks, developing a robust evaluation methodology that achieves an 84.4% success rate in determining the acceptability of generated responses.

Our findings highlight both the strengths and limitations of current LLMs in coding assistance tasks. The StackEval benchmark reveals that LLMs perform exceptionally well on historical content and common programming tasks. However, as questions become more complex and niche, their accuracy notably decreases. The StackUnseen benchmark further demonstrates the challenges LLMs face in generalization. When presented with emerging technologies and recent coding practices, LLMs exhibit a significant drop in performance compared to their results on more established programming paradigms. This performance gap illustrates the adaptability limitations of current LLMs to rapidly evolving coding landscapes.

**Ethical Considerations.** The widespread adoption of LLM-based coding presents significant ethical considerations for software engineering. Our benchmarks demonstrate that top-performing models achieve high acceptance rates for various coding tasks, potentially altering the job market for software developers. This raises concerns about diminished opportunities for junior developers to gain hands-on experience and develop crucial problem-solving skills, which could contribute to a decline in overall code quality over time.

LLMs, trained on publicly available code repositories with varying quality, may perpetuate or exacerbate existing code quality issues due to the limited availability of exemplary code in these datasets. Our LLM-as-a-Judge benchmark reveals that even the best models fall short of perfect accuracy in evaluating code solutions, highlighting the continued necessity for human oversight, especially in critical systems.

There is also a risk of perpetuating biases present in the training data, potentially leading to unfair or discriminatory outcomes. As the field progresses, it is crucial to develop new standards for code review, testing, and accountability that balance AI assistance with human expertise, while continuously monitoring and addressing these issues to ensure positive technological advancement.

# 8   Acknowledgments

We thank the Prosus AI team for their help in labeling the LLM-as-a-Judge dataset and our colleagues at Stack Overflow for providing the data.

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

# A   Appendix

| Model | Provider | StackEval | StackUnseen | LLM-as-a-Judge |
|---|---|---|---|---|
| O1 Preview | OpenAI | **95.5%** | **83.0%** | 80.9% |
| O1 Mini | OpenAI | 93.8% | 80.9% | 78.7% |
| Gemini-1.5 Pro | Google | 90.7% | 71.6% | 76.4% |
| Claude-3.5 Sonnet | Anthropic | 89.5% | 76.3% | 83.0% |
| GPT-4 Turbo | OpenAI | 88.0% | 61.3% | **86.2%** |
| Llama3.1-Nemotron-70B | Nvidia | 86.9% | 66.5% | 76.5% |
| Gemini-1.5 Flash | Google | 83.4% | 63.4% | 75.7% |
| GPT-4o | OpenAI | 83.0% | 55.7% | 82.3% |
| Mistral Large 2 | Mistral | 81.9% | 52.6% | 80.7% |
| Qwen2.5-72B-Instruct | Alibaba | 80.4% | 50.5% | 80.1% |
| WizardLM-2 8x22B | Microsoft | 80.2% | 50.5% | 81.8% |
| GPT-4o Mini | OpenAI | 80.0% | 42.3% | 81.6% |
| Deepseek Coder-v2.5 | DeepSeek AI | 78.7% | 43.3% | 80.9% |
| Llama3.1-405B Instruct | Meta | 76.5% | 47.9% | 78.7% |
| Claude-v3 Opus | Anthropic | 75.0% | 47.4% | 76.5% |
| GPT-4 | OpenAI | 74.2% | 42.3% | 82.4% |
| Llama3.1-70B Instruct | Meta | 73.5% | 41.2% | 82.1% |
| Mistral Small | Mistral | 67.0% | 34.5% | 77.2% |
| Gemma-2-27B Instruct | Google | 63.6% | 38.7% | 77.2% |
| Gemma-2-9B Instruct | Google | 62.9% | 36.1% | 68.4% |
| Claude-v3 Haiku | Anthropic | 60.8% | 32.5% | 66.2% |
| GPT-3.5 Turbo | OpenAI | 54.4% | 22.2% | 66.9% |
| Llama3.1-8B Instruct | Meta | 54.1% | 17.0% | 66.2% |
| Mistral Nemo | Mistral | 43.4% | 14.9% | 63.2% |

Table 5: **StackEval, StackUnseen and LLM-as-a-Judge Benchmark.** Performance comparison between various LLMs [21, 33, 3, 22, 20, 23, 1, 35, 38, 11, 12, 4, 34, 2] on coding assistance benchmarks (StackEval & StackUnseen), alongside the LLM-as-a-Judge benchmark (CoT + Ref. Answer).

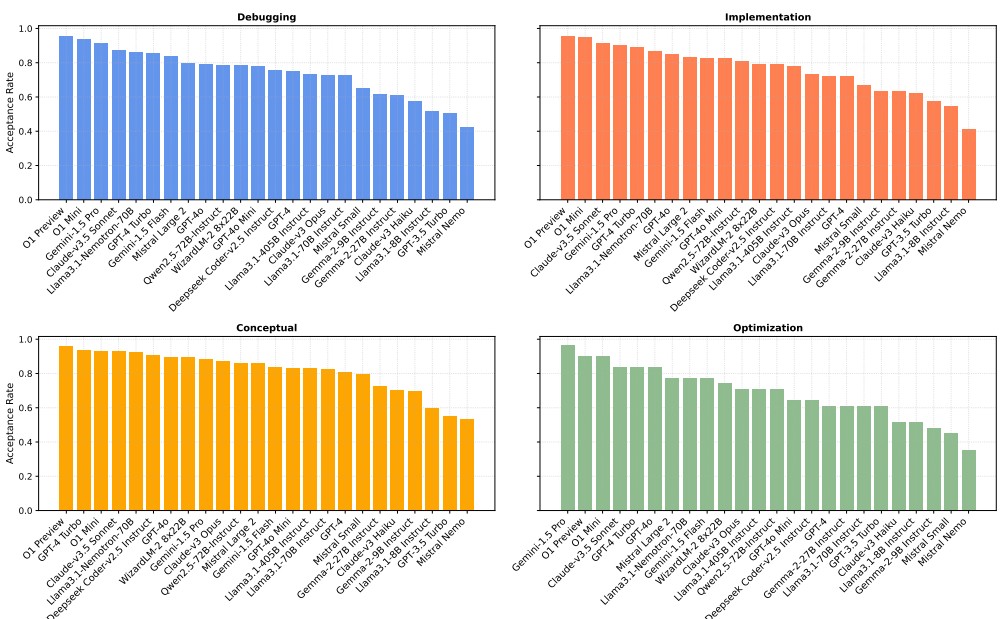

Figure 5: The performance of various LLMs across different question types on the **StackEval benchmark**, averaged across programming languages.

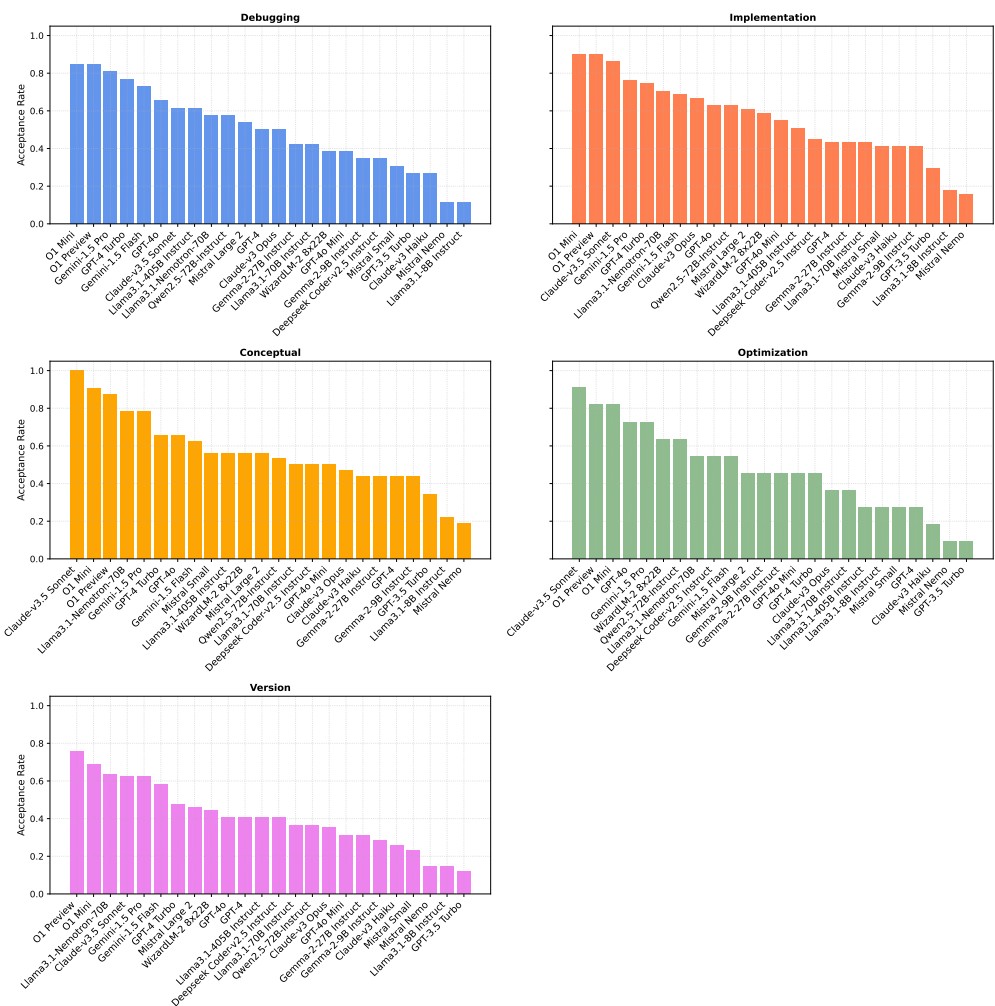

Figure 6: The performance of various LLMs across different question types on the **StackUnseen benchmark**, averaged across programming languages.

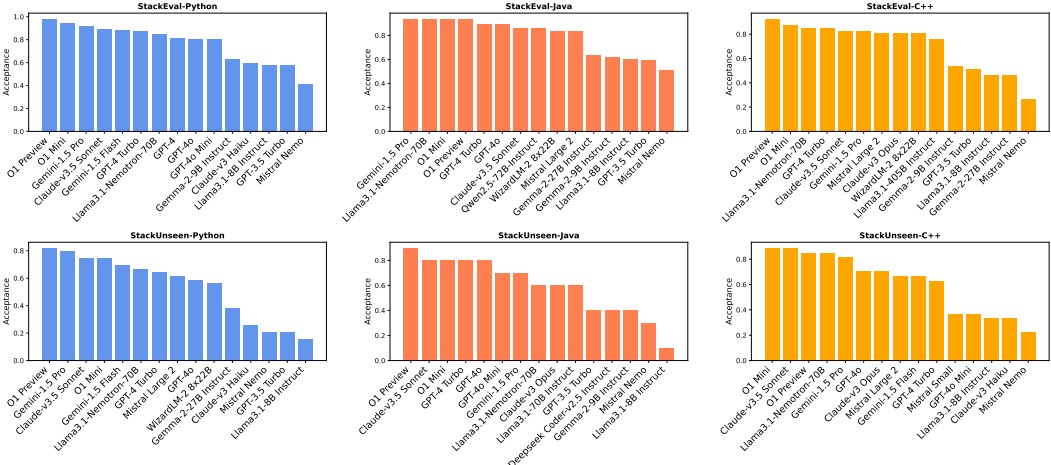

Figure 7: The performance of various LLMs across popular programming languages on the **StackEval** and **StackUnseen** benchmark.

## LLM Evaluation Prompt

**Task Description** You are a very experienced and knowledgeable answer checker. You will be given a question, a reference answer and an LLM generated answer. Your task is to evaluate how good the answer is in answering the question of the user. More specifically, you will evaluate the acceptability of the answer for the user following the definition and rubric below.

**Acceptability Definition**

Acceptability measures how effectively an answer satisfies the user's specific requirements and addresses their issue. It evaluates whether the response provides a viable solution, focusing on the answer's accuracy, relevance, and completeness. An acceptable answer:
- Is one that the user would regard as a fitting resolution to their query
- Enables the user to proceed without requiring additional help or verification
- May not be perfect and may contain small inaccuracies that will not affect usability
- Must work without user editing (for code) or cover crucial points (for advice)

**Acceptability Evaluation Rubric**

**Score 0 – Completely Unacceptable** • Incorrect or entirely irrelevant, with substantial errors
- Contains severe hallucinations or misinformation, significantly misleading the user
- Leaves significant gaps, necessitating further search for information
- User would immediately disregard this answer

**Score 1 – Useful but Unacceptable** • Contains some correct information but significant inaccuracies or lacks important details, prompting additional research
- Somewhat relevant but misses critical nuances, leading to an incomplete understanding
- Not comprehensive, omitting important aspects and critical details needed to solve the user's problem
- Provides some value but requires further searching for a complete and satisfactory solution

**Score 2 – Acceptable** • Accurate, free of critical errors that would prevent problem resolution
- Relevant and demonstrates a clear understanding of the issue, addressing the main points and considerations, and directly applicable to the problem.
- Offering a satisfactory solution, even if it is not the most optimal solution. Minor details may be omitted, but nothing vital is missing.
- Provides enough information for the user to proceed without additional help

**Score 3 – Optimal** • The answer is 100% accurate and provides a detailed response, where the details improve answers quality and usability
- It is thorough and addresses additional relevant aspects that could enhance the user's understanding of the solution.
- The response may include extra information, such as best practices or helpful tips, that adds value and could assist the user in avoiding common mistakes or in understanding the broader context.
- The user is likely to feel well-informed and be able to apply the solution effectively

**Threshold Definition** The critical threshold between Score 1 and Score 2:
- Score 1 (Useful but Unacceptable): Provides correct information but requires additional research.
- Score 2 (Acceptable): Offers complete, accurate information allowing issue resolution without further resources.

**Assessment Process**
1. Analyze question and reference answer for core requirements.
2. Evaluate generated answer against requirements and reference.
3. Reason on the acceptability of the generated answer based on the definition.
4. Assign final score based on rubric.

**Output Format** The evaluation should be formatted as a JSON object:

```
{
  "questionAnalysis": "Review core elements required for answer",
  "generatedAnswerAnalysis": "Evaluate coverage, strengths, and weaknesses",
  "acceptabilityEvaluation": "Assess accuracy, relevance, and completeness",
  "acceptabilityScore": "Assign the most appropriate score, <int: 0-3>"
}
```

Figure 8: The LLM evaluation prompt (CoT + Ref. Answer) used to assess answer quality.

I need to know the exact number of arguments that a lambda has. I do not care for their types, I just need a count.

```cpp
auto lambda0 = [&]() { ... };
auto lambda1 = [&](int32_t a) { ... };
auto lambda2 = [&](int32_t a, auto b) { ... };

lambda_details<decltype(lambda0)>::argument_count; // Equals 0
lambda_details<decltype(lambda1)>::argument_count; // Equals 1
lambda_details<decltype(lambda2)>::argument_count; // Equals 2
```

Detecting variadic lambdas would also be nice so that I can deal with that edge case as well.

```cpp
auto lambda_variadic = [&](auto... args){ ... };

lambda_details<decltype(lambda_variadic)>::is_variadic; // Equals true
```

How can I get this information?

Figure 9: **StackEval Implementation.** A C++ implementation sample question from the StackEval dataset.

Call to undefined method Illuminate/Routing/RouteFileRegistrar::get() - Error after upgrading from Laravel 5.7 to 5.8. I have a running app written on Laravel 5.7. I tried to change the record in composer.json to match "5.8.*" and ran composer update. On my local (win10/WAMP) machine it went fine, but on the staging server (Debian 9/nginx) the update command changed the vendor contents and failed at the end.
Since then anything I do with the app on the server I get this error and I can't find any information anywhere.

```
Call to undefined method Illuminate\Routing\RouteFileRegistrar::get()
```

And this is the line that fails:

```
$this->get('login', 'Auth\LoginController@showLoginForm')->name('login');
```

Figure 10: **StackEval Debugging.** A PHP debugging sample question from the StackEval dataset.

Accessing something inside the object when you don't know the key. I am getting a following object:

```
{
    IuW1zvaSABwH4q: {
        label: 'Random Image of TypeScript not relavent to coworking',
        thumbId: 'd501-f-b601-c8b1-4bd995e',
        schemaType: 'xman-assets-image-set'
    }
}
```

Now, I want to access the value of thumbID inside it i.e. `d501-f-b601-c8b1-4bd995e`, but my root key seems to be dynamic/random (`IuW1zvaSABwH4q`). How can I access the value inside it?

Figure 11: **StackEval Conceptual.** A conceptual style sample question from the StackEval dataset.

I have a vector of prices (f64). I would like to compute the highest price. What is the current easiest and most idiomatic way to compute the max of a collection of f64 in rust?
There has been some discussion about Ord and f64 but I am not sure what is the most up-to-date and less hacky way to do so.
I rely on the following but I imagined there was some built in operation:

```
let max = prices.iter().fold(None, |r, &n| match r {
    Some(p) => Some(f64::max(p, n)),
    None => Some(e),
});
```

(which is just a fold for some free monoid)

Figure 12: **StackEval Optimization.** A Rust optimization question from the StackEval dataset.

Why ngModel doesn't works on the last version of Angular 17? I am trying to make a form in my angular app, but when i want to implement ngModel on my form:

```
<form (ngSubmit)="onSignUp()" #signupForm="ngForm">
    <h1>Connexion</h1>
    <input
        type="email"
        name="mail"
        [(ngModel)]="userLogin.email"
        placeholder="Email"
    />
    <input
        type="password"
        name="mdp"
        [(ngModel)]="userLogin.password"
        placeholder="Password"
    />
    <a href="#">Mot de passe oublie ?</a>
    <button type="submit">Se Connecter</button>
</form>
```

I have this error: `NG8002:  Can't bind to 'ngModel' since it isn't a known property of 'input'.  [plugin angular-compiler]`. I can't import FormsModule in the app.module.ts because this file doesn't exists on Angular 17, i only have an app.config.ts file. Can someone please explain me how to do?

Figure 13: **StackUnseen Versioning.** A version-dependent question from the StackUnseen dataset.

