# OpenReview forum: "StackEval: Benchmarking LLMs in Coding Assistance"
_NeurIPS.cc/2024/Datasets_and_Benchmarks_Track — NeurIPS 2024 Track Datasets and Benchmarks Poster_

### Official Review · Reviewer_nCkZ · 2024-07-18
**Interesting dataset for a highly relvant problem; some concerns about the internal validity.**

**Rating:** 7
**Confidence:** 4
**Clarity:** The paper is well written and underst…

**Review:**

I really like the paper and the approach - especially the idea of having unseen data and keeping this up-to date is really good.

However, I am a bit unsure about the validity of the approach. The authors use existing LLMs in creating the data set in two places:
1) they annotate questions with a question type and a complexity level (line 117)
2) LLMs are used to create the LLM-as-a-judge benchmark (line 145)

For the first point I am missing a justification why this works. So is the type and complexity assessment of chatgpt good and how good is this? I would expect some comparison to human assessments of a subset of the data. Without knowing this, it might invalidate dataset as it would no longer be balanced regarding type and complexity.

For the second point the authors say that the questions are validated by domain experts. Yet I fear that the LLM will reproduce its training set thus the answers to the questions will be part of the system's training set. If this is true the LLM-as-a-judge dataset cannot be used to test the quality of LLMs, as at least the LLMs used to generate the dataset will have an advantage.

**Strengths:**

The dataset, especially the two curated SO-dataset are highly useful for future research on this topic.

**Additional Feedback:**

In Figure 1 the authors could indicate in the bars in different color the types of questions that are in the dataset.

**Correctness:**

Aside from the impact of LLMs on the dataset, I believe the paper to be correct.

**Documentation:**

I was able to use the dataset and open the JSON files, this is sufficient.

**Ethics:**

None aside from the weak ethics section.

**Limitations:**

Here again, I feel that the impact of LLMs on the dataset is not considered clearly enough. Unfortunately the paper is missing a Limitations section. Additionally, the paper provides a short paragraph on the ethical limitation, but lacks citations or any in-depth connection to the content of the paper. This section should  be improved.

**Opportunities For Improvement:**

I feel that the paper would be greatly improved with a more in-depth reflection on the use of LLMs in the creation of the dataset.

**Relation To Prior Work:**

Prior work is mentioned, however missing in the ethical reflections.

**Summary And Contributions:**

The authors provide a dataset that is comprised of three distinct data: Old stack-overflow questions, fairly recent stack-overflow questions that are probably not part of any training set and LLM-generated questions that are probably unseen by current LLMs.

---

> ### Author Rebuttal · Authors · 2024-08-17
>
> Dear Reviewer nCKz,
>
> Thank you for your valuable feedback on our paper. We appreciate your insights and plan to address your concerns as follows:
>
> ### **1. The `question-type` and `complexity` annotations**
> To address your concern, we are in the processes of conducting the following experiment:
> - Sample 50 questions and obtain human annotations to verify accuracy of LLM-generated annotations.
> - Calculate the agreement rate between human and LLM annotations.
>
> We will report the findings as soon as possible and include them in the revised paper. Additionally, we'd like to clarify that the `question-type` and `complexity` metadata was not used for sampling but rather as a descriptive field. The sampling was done according to other, more objective filters like total upvote of the question/answer. We acknowledge that the complexity annotation is subjective and can vary from person to person. In light of this, we will present it as a supplementary indicator that may be helpful but should not be considered definitive.
>
> ### **2. Explanation of LLM-as-Judge Benchmark**
> Indeed, our explanation of the LLM-as-Judge benchmark could have been clearer. We have attached a flow diagram to better illustrate the process. To clarify:
>
> The LLM-as-a-Judge benchmark aims to evaluate LLMs' capability in assessing the quality of other LLMs' answers to Stack Overflow questions. We curated 135 diverse, high-quality questions from Stack Overflow, covering various programming tasks and languages. For each question, we generated an answer using one of four LLMs: CodeLlama 34B (31 answers), GPT-3.5 (29 answers), GPT-4-1106 (38 answers), and Mistral Medium (37 answers). This diverse set ensures a comprehensive evaluation across different model capabilities. We also included the accepted Stack Overflow answer as the ground truth for each question.
>
> In our evaluation, we presented each LLM judge with a tuple containing the original question, the ground truth answer, and an LLM-generated answer. The LLM was then asked to evaluate the quality of the LLM-generated answer. To ensure accuracy, we employed four distinct methods: (1) providing only the question and answer, (2) including a reference answer, (3) using chain-of-thought reasoning, and (4) combining reference answers with chain-of-thought reasoning. We ran each evaluation 10 times to account for response variability. Simultaneously, we presented the same LLM-generated answers to three domain experts for labeling, with an additional expert validating scores in cases of disagreement. We then calculated the accuracy of the LLM evaluations by comparing their acceptability scores to the final scores from the domain experts.
>
> Additionally:
> - `LLM-t`’s (see flow diagram) familiarity with StackOverflow questions doesn't impact our evaluation as we provide the complete tuple (StackOverflow question, StackOverflow answer (as ground-truth), LLM-x answer) for each assessment. Thus each judge gains familiarity before evaluation.
> - This tuple is inherently out of distribution for any pre-trained LLM, due to the inclusion of `LLM-x` answer, mitigating train-test overlap concerns.
>
> However, we acknowledge a valid limitation: LLMs may prefer their own generations [[1]](https://arxiv.org/abs/2404.13076). To address this concern and provide further insights, we are conducting an additional experiment:
> - The LLM-as-Judge benchmark contains LLM answers generated by one of the four LLMs: CodeLlama-34B (31 answers), GPT-3.5 (29 answers), GPT-4 Turbo (38 answers) and Mistral Medium (37 answers).
> - We will then use each of these four LLMs to evaluate the answers generated by all four LLMs, and report the individual evaluation accuracies.
>
> This will help us identify any potential bias an LLM might exhibit towards answers generated by models within its own family. We will include this additional information in the revised paper.
>
> ### **3. Ethics section**
> We appreciate your feedback on the ethics section. For the revised paper, we shall incorporate additional citations providing a more comprehensive discussion of the ethical implications of our work.
>
> ### **4. Visualization Improvement**
> We will update our visualizations to indicate the types of questions in the dataset. Additionally, we shall add flow-diagrams for the different processes (data curation, LLM evaluations, etc.) to make our methodology clearer.
>
> We hope we have addressed your concerns adequately. Thank you for your valuable input!

---

> > ### Comment · Reviewer_nCkZ · 2024-08-29
> >
> > Thank you for your thorough answers - I will raise my score to accept.

---

> > > ### Author Response · Authors · 2024-08-29
> > >
> > > We humbly thank you for raising your rating of our work! All the relevant points will be incorporated into the final version of the paper.

---

### Official Review · Reviewer_1ibX · 2024-07-23
**Useful Benchmarks for LLM for Code Community**

**Rating:** 6
**Confidence:** 4
**Clarity:** The paper was easy to read.

**Review:**

Quality: the benchmarks are comprehensive and diverse in terms of the tasks and languages they evaluate, and robustness of evaluations.

Clarity: paper is well written and easy to read.

Originality: the Related Works section was simple, but to the best of my knowledge, this is the first paper that:
- evaluate LLM's performance on recent questions/concepts.
- evaluates LLM's performance in judging.

Significance: while the insights in the Results section are not surprising, overall I belief the benchmarks are useful for researchers to evaluate thier benchmark.

**Strengths:**

- Diverse set of question types and languages
- Comprehensive evaluation of LLM-as-a-Judge
- Comprehensive evaluations with detailed Scores, and experiments
- Detailed results

**Additional Feedback:**

Line 227: The acronym CA-R seems to be specified for the first time here. Please clarify what it stands for.

**Correctness:**

- Datasets seem to be constructed in a sound way
- Evaluation methods and experiment design seem to be correct. StackEval and StackEval-Recent were evaluated by LLMs. However, LLM evaluation capabilities were tested by humans.

**Documentation:**

- Data collection and organization: was explained (based on StackOverflow)
- Availability: open sourced
- Maintenance: authors are committed to update StackEval-Recent with updated datasets

**Limitations:**

- I am concerned that StackEval and StackEval-Recent is only evaluated by LLMs. I would have preferred if the questions/answers could be evaluated in small self-contained unit tests that could be evaluated in runtime (e.g., questions about Debugging test the correctness of the generated code with the fixed bug, while questions about Optimizations measure both correctness and speed of generated code). This would make the benchmark more robust and reliable. This would indeed require more effort in tailoring or re-writing the StackOverflow questions into self-contained code.

- Although effort was put in filtering StackOverflow questions, effort wasn't put to verify answers of LLM models other than evaluating it using another LLM model. For example, benchmarks like [1], HumanEval and MBPP do test proposed code changes by executing them in runtime or socialism.

[1] "Learning Performance-Improving Code Edits", Alexander G Shypula, Aman Madaan, Yimeng Zeng, Uri Alon, Jacob R. Gardner, Yiming Yang, Milad Hashemi, Graham Neubig, Parthasarathy Ranganathan, Osbert Bastani, Amir Yazdanbakhsh, ICLR 2024

**Opportunities For Improvement:**

- Related Work section could have been more borad.
- Add a Figure that summarizes the different categories and benchmarks in StackEval. Perhaps showing a short trivial example for each category for illustrative purposes
- Provide number of questions that fall under each of the categories: Conceptual, Debugging, Implementation, and Optimization
- Provide 1 or 2 samples in the main body of the paper or in the Appendix for each of the categories: Conceptual, Debugging, Implementation, and Optimization
- Provide model performance under each of the categories: Conceptual, Debugging, Implementation, and Optimization
- Provide pairwise correlation of model's performance between:
   - each 2 categories,
   - between StackEval and StackEval-Recent
   - between StackEval and HumanEval or MBPP
- Integrating the code to evaluate the benchmark into HuggingFace and BigCode evaluation will enable widespread use of the benchmarks

**Relation To Prior Work:**

- Related Work section could have been more borad. For example, it should have mentioned other papers that evaluated LLM's capabilities to optimize code (e.g.,[1])

[1] "Learning Performance-Improving Code Edits", Alexander G Shypula, Aman Madaan, Yimeng Zeng, Uri Alon, Jacob R. Gardner, Yiming Yang, Milad Hashemi, Graham Neubig, Parthasarathy Ranganathan, Osbert Bastani, Amir Yazdanbakhsh, ICLR 2024

**Summary And Contributions:**

This paper provides StackEval: a suite of becnhmarks that evaluate LLMs in answering different types of questions: Conceptual, Debugging, Implementation, and Optimization. It includes StackEval benchmark, as well as StackEval-Recent benchmark to evaluate model's capabilities on recent questions or concepts, and LLM-as-a-Judge Benchmark to evaluate LLM's capabilities to judge between provided answers.

The paper provides analysis on different models and shows interesting insights and correlations.

---

> ### Author Rebuttal · Authors · 2024-08-17
>
> Dear Reviewer 1ibX,
>
> Thank you for your feedback. We plan to address your concerns as follows:
>
> ### **1. Providing examples for each category**
> We agree that illustrative examples would enhance the clarity of our work. We will add sample questions and answers for each category (conceptual, debugging, implementation and optimization) in the appendix.
>
> ### **2. Model performance under each category**.
> Thank you for this suggestion, we agree that it’d be very useful to add this. We will include additional tables in the revised paper showing model performance for each of the categories.
>
> ### **3. Pair-wise correlation between benchmarks and question-types**
> We have conducted this analysis and included the pair-wise correlation between StackEval and StackEval-Recent (renamed to StackUnseen for clarity) as well as the different question-types. We have also included correlation with HumanEval and LMSYS ChatBotArena to provide a more comprehensive comparison. We will include the additional material in our paper.
>
> ### **4. Verification of LLM answers and lack of unit-tests**
> Our StackEval benchmarks are positioned between technical Q&A and code-completion benchmarks, reflecting real-world LLM use in coding scenarios. Given the diverse and frequently updated nature of our question set (especially StackUnseen), implementing unit tests for each question would probably not be feasible. Instead, we've developed an LLM-as-judge approach which we believe to be robust due to the following validation process:
> - We conducted extensive labeling efforts with multiple domain experts.
> - Our process involved three rounds of labeling, with additional experts brought in to resolve disagreements.
> - We achieved 85% agreement between our LLM-as-judge and human annotations.
>
> This approach balances answer quality verification with the need for a large-scale, updatable benchmark. We acknowledge the limitations of this method as you also have outlined and will discuss them in our paper, following your suggestion
>
> For future work, we're exploring an agent-based LLM system equipped with a code-executor tool. This system could dynamically write unit tests during evaluation and execute the proposed LLM solutions, potentially further improving accuracy. We believe this represents the next step in enhancing our validation process.
>
> ### **5. Typo: CA-R**
> We apologise for this inconsistency. This was intended to reference to our StackEval-Recent dataset. To improve clarity, we will rename StackEval-Recent to StackUnseen throughout the paper and fix all the references.
>
> Thank you again for your insightful comments that helped improve our paper.

---

### Official Review · Reviewer_KPZb · 2024-07-25
**A Comprehensive Evaluation of LLMs for Coding Assistance**

**Rating:** 8
**Confidence:** 3

**Review:**

The paper "StackEval: Benchmarking LLMs in Coding Assistance" presents a rather ambitious and comprehensive approach to evaluating Large Language Models in the domain of coding assistance. The authors have clearly put a great deal of thought and effort into developing a suite of benchmarks that address several key limitations in existing evaluation frameworks.

The StackEval benchmark, with its impressive coverage of 25 programming languages and 925 curated questions, offers a substantial improvement over existing benchmarks in terms of scope and diversity. This breadth allows for a more nuanced understanding of LLM performance across different programming paradigms and task types, which is jolly good for the field.

The introduction of StackEval-Recent is particularly commendable. By focusing on the most recent Stack Overflow content, this benchmark addresses the critical issue of evaluating LLM performance on new and emerging programming challenges. This approach helps mitigate the risk of benchmark leakage and provides valuable insights into the adaptability of LLMs to evolving coding practices.
The LLM-as-a-Judge benchmark is an innovative addition that explores the potential of using LLMs themselves as evaluators. This could prove quite useful in developing more scalable and consistent evaluation methodologies for open-ended tasks in coding assistance.

The enhanced auto-evaluation methodology, achieving an 85% success rate in assessing the effectiveness of LLMs in judging the acceptability of generated responses, is a promising step towards more reliable automated evaluations.
However, while the paper presents several strengths, there are areas where improvements could be made:

- The paper would benefit from a more detailed discussion of the potential biases inherent in using Stack Overflow as the primary source of questions. While Stack Overflow is undoubtedly a rich resource, it may not fully represent the diversity of coding challenges faced in various professional contexts.
- The authors could provide more information on how they ensured the quality and reliability of the human expert annotations used in the LLM-as-a-Judge benchmark. A more detailed description of the annotation process and inter-annotator agreement would strengthen the credibility of this benchmark.

Overall, this paper represents a valuable contribution to the field of LLM evaluation in coding assistance. The comprehensive nature of the benchmarks and the thoughtful approach to addressing existing limitations in the field are commendable. With some refinements and additional details, this work has the potential to significantly impact how we evaluate and understand the capabilities of LLMs in coding tasks.

**Strengths:**

- Comprehensive coverage of 25 programming languages, providing a diverse and representative benchmark for coding assistance tasks.
- Introduction of StackEval-Recent, addressing the challenge of evaluating LLM performance on new and emerging coding challenges.
- Innovative LLM-as-a-Judge benchmark, exploring the potential of using LLMs for evaluation tasks.
- Enhanced auto-evaluation methodology with a high success rate, contributing to more reliable automated assessments.
- Clear commitment to ongoing updates and maintenance of the benchmarks, ensuring their continued relevance.
- Thoughtful approach to mitigating benchmark leakage and exposure bias.

**Additional Feedback:**

(blank)

**Clarity:**

The paper is generally well-written and organized logically. The authors have done a good job of explaining complex concepts and methodologies in a clear and accessible manner. However, some sections, particularly those detailing the evaluation methodology, could benefit from additional examples or visual aids to enhance understanding.

**Correctness:**

The paper is generally well-written and organized logically. The authors have done a good job of explaining complex concepts and methodologies in a clear and accessible manner. However, some sections, particularly those detailing the evaluation methodology, could benefit from additional examples or visual aids to enhance understanding.

**Documentation:**

The paper provides sufficient detail on data collection, organization, and evaluation methods. The authors' commitment to making the StackEval and StackEval-Recent datasets publicly available, with plans for regular updates, is praiseworthy. However, more information on the specific hosting, licensing, and maintenance plans would be beneficial.

**Ethics:**

The authors have demonstrated awareness of potential ethical concerns, particularly regarding bias perpetuation and job displacement. However, there are a few areas that warrant further discussion:

- Data privacy and consent: While the use of publicly available Stack Overflow data mitigates some concerns, a more detailed discussion of how user privacy is protected in the dataset would be beneficial.
- Representativeness: A deeper exploration of potential biases in the Stack Overflow data and how these might affect the benchmark's representativeness across different demographics and geographic regions would strengthen the paper.
- Potential for misuse: The authors could expand on the potential for these benchmarks to be misused, such as over-optimizing LLMs for benchmark performance at the expense of real-world applicability.

In conclusion, while the paper addresses several ethical considerations, a more comprehensive discussion of these issues would further enhance its contribution to responsible AI development in the field of coding assistance.

**Limitations:**

The authors have made a commendable effort to address the limitations of their work, particularly in acknowledging the potential for perpetuating biases present in the training data and the risk of job displacement in the software development industry. However, there are opportunities for a more comprehensive discussion:

- The paper could benefit from a more detailed analysis of the potential biases inherent in Stack Overflow data, including demographic and geographic biases that might affect the representativeness of the benchmarks.
- A more thorough exploration of the limitations of using LLMs as judges, including potential biases and limitations in their evaluation capabilities, would strengthen the paper.
- The authors could provide a more in-depth discussion of the potential long-term impacts of increased reliance on LLM-based coding assistance on software engineering education and skill development.

**Opportunities For Improvement:**

Opportunities for Improvement:

- Deeper analysis of potential biases in using Stack Overflow as the primary source of questions.
- More detailed information on the human expert annotation process for the LLM-as-a-Judge benchmark.
- Clearer strategy for long-term maintenance and updating of the benchmarks.
- Expanded discussion of ethical considerations and potential negative societal impacts.
- More extensive comparison with existing benchmarks to highlight the specific advancements made by StackEval.
- Inclusion of case studies or specific examples demonstrating how these benchmarks reveal new insights about LLM capabilities in coding assistance.

**Relation To Prior Work:**

The authors have done a commendable job of discussing how this work differs from previous contributions. They provide a comprehensive overview of existing coding benchmarks and evaluation methodologies for LLMs, clearly highlighting the limitations of current approaches and how StackEval addresses these gaps.

**Summary And Contributions:**

This paper presents StackEval, StackEval-Recent, and LLM-as-a-Judge, three novel benchmarks designed to evaluate Large Language Models (LLMs) in coding assistance tasks. The primary contributions include:

- A comprehensive coding assistance benchmark (StackEval) with 925 Stack Overflow questions across 25 programming languages.
- A recent coding assistance benchmark (StackEval-Recent) with 300 questions from the most recent Stack Overflow content.
- An LLM-as-a-Judge benchmark featuring 136 LLM-generated answers validated by domain experts.
- An enhanced auto-evaluation methodology for assessing LLM performance in coding tasks.

The authors aim to provide a more robust and up-to-date evaluation framework for LLMs in the context of coding assistance, addressing limitations in existing benchmarks and offering insights into LLM capabilities across various programming languages and task types.

---

> ### Author Rebuttal · Authors · 2024-08-17
>
> Dear Reviewer KPZb,
>
> We sincerely appreciate your thorough review and the valuable feedback you've provided. We're pleased that you found merit in our work, as reflected in your high score of 8. We've carefully considered your suggestions and would like to address them as follows:
>
> ### **1. Biases in StackOverflow Data**
> We agree that this is an important issue to address. We will add a deeper analysis of this issue in the appendix. Specifically, we shall discuss how StackOverflow's rules for posting questions and answers might affect the distribution of questions in our dataset, as well as the ban on LLM-generated answers/questions. We will include a reference to StackOverflow's meta-website, which outlines these guidelines.
>
> ### **2. Long-term Maintenance of Benchmarks**
> We'd like to clarify that upon acceptance, we shall host the StackEval benchmark and make it publicly available via mediums such as GitHub. For StackUnseen (previously referred to in the paper as StackEval-Recent), we plan to have a quarterly refresh cycle. In fact, we've already obtained a new batch of questions and evaluated the models on this updated set.
>
> ### **3. Regarding the long-term impact of increased reliance on LLM-based Coding Assistance**
> We will include a discussion on this topic to address the potential effects on software engineering education and skill development. We shall highlight several key-points:
> - The distribution of the code the LLMs are trained on, and how that would affect their output.
> - LLMs' ability to handle beginner-level questions, potentially freeing up humans moderators to focus on more complex, expert-level inquiries.
> - We discuss StackOverflow's ban on LLM-generate answers, contextualising the importance of human expertise.
>
> ### **4. User Privacy Protection in the dataset**
> We shall clarify in the revised paper that we only retain the questions themselves, without any user identifiers such as usernames. Furthermore, we emphasise that these questions were originally posted on a public forum, making them open for use in research contexts like ours.
>
> ### **5. Over-optimizing LLMs for benchmark performance**
> We appreciate this concern and would like to emphasise on how StackUnseen helps mitigate this risk. By regularly updating with new, unseen questions, we ensure that the benchmark remains challenging and representative of real-world coding scenarios.
>
> Once again, we would like to express our gratitude for your encouraging score of 8. Your sunlights have helped us significantly improve our paper.

---

### Decision · Program_Chairs · 2024-09-26

**Decision:**

Accept (Poster)

**Comment:**

Thanks for your submission to NeurIPS 2024. Overall, the reviewers recognize that the paper presents a well-constructed and novel benchmark for evaluating the coding assistance abilities of LLMs among different downstream tasks. In light of this, the reviewers agree that this paper makes important contributions to the community and is valuable to be accepted.

However, please carefully address the reviewers' comments in the camera-ready version. In particular, consider including additional discussion about the potential biases introduced by data and the limitations of using LLMs as judges.